# VARIATIONAL DETERMINISTIC UNCERTAINTY QUANTIFICATION

## ABSTRACT

Building on recent advances in uncertainty quantification using a single deep deterministic model (DUQ), we introduce variational Deterministic Uncertainty Quantification (vDUQ). We overcome several shortcomings of DUQ by recasting it as a Gaussian process (GP) approximation. Our principled approximation is based on an inducing point GP in combination with Deep Kernel Learning. This enables vDUQ to use rigorous probabilistic foundations, and work not only on classification but also on regression problems. We avoid uncertainty collapse away from the training data by regularizing the spectral norm of the deep feature extractor. Our method matches SotA accuracy, 96.2% on CIFAR-10, while maintaining the speed of softmax models, and provides uncertainty estimates competitive with Deep Ensembles. We demonstrate our method in regression problems and by estimating uncertainty in causal inference for personalized medicine.

## 1 INTRODUCTION

Deploying machine learning algorithms as part of automated decision making systems, such as self driving cars and medical diagnostics, requires implementing fail-safes. Whenever the model is presented with a novel or ambiguous situation, it would not be wise to simply trust its prediction. Instead, the system should try to get more information or simply withhold or defer judgment. While significant progress has been made towards estimating predictive uncertainty reliably in deep learning (Gal & Ghahramani, 2016; Lakshminarayanan et al., 2017), there is no single method that is shown to work on large datasets in classification and regression without significant computation overheads, such as multiple forward passes. We propose Variational Deterministic Uncertainty Quantification (vDUQ), a method for obtaining predictive uncertainty in deep learning for both classification and regression problems in only a single forward pass.

In previous work, van Amersfoort et al. (2020) show that combining a distance aware decision function with a regularized feature extractor in the form of a deep RBF network, leads to a model (DUQ) that matches a softmax model in accuracy, but is competitive with Deep Ensembles for uncertainty on large datasets. The feature extractor is regularized using a two-sided gradient penalty, which encourages the model to be sensitive to changes in the input, avoiding feature collapse, and encouraging generalization by controlling the Lipschitz constant. This model, however, has several limitations; for example the uncertainty (a distance in feature space) cannot be interpreted probabilistically and it is difficult to disentangle aleatoric and epistemic uncertainty. Additionally, the loss function and centroid update scheme are not principled and do not extend to regression tasks.

A probabilistic and principled alternative to deep RBF networks are Gaussian Processes (GPs) in combination with Deep Kernel Learning (DKL) (Hinton & Salakhutdinov, 2008; Wilson et al., 2016b). DKL was introduced as a "best of both worlds" solution: apply a deep model on the training data and learn the GP in feature space, ideally getting the advantages of both models. In practice, however, DKL suffers from the same failure as Deep RBF networks: the deep model is free to map out of distribution data close to the feature representation of the training data, removing the attractive properties of GPs with distance sensitive kernels.

Using insights from DUQ, we are able to mitigate the problems of uncertainty collapse in DKL. In particular, we use direct spectral normalization (Gouk et al., 2018; Miyato et al., 2018) in combination with a ResNet (He et al., 2016), a variation that was suggested in Liu et al. (2020). The spectral

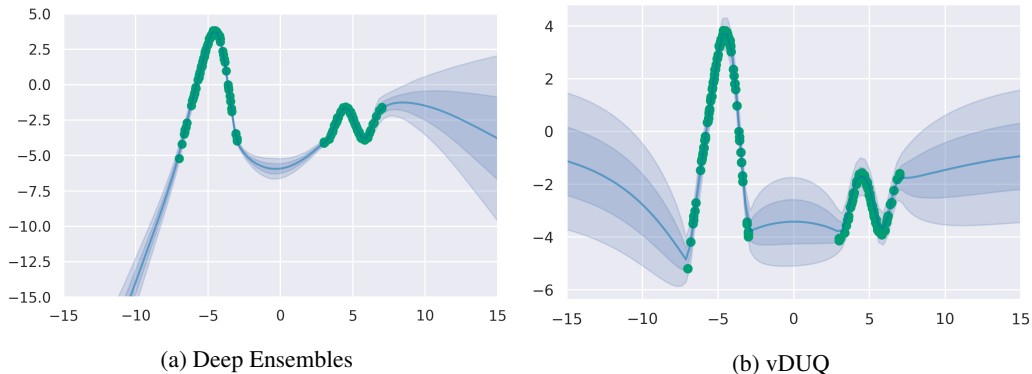

(a) Deep Ensembles                                    (b) vDUQ

Figure 1: We show results on a 1D regression dataset of a sinusoid curve. In green are data points of the dataset and in blue the prediction including uncertainty (two standard deviations). As expected, vDUQ reverts to the prior away from the training data, while Deep Ensembles extrapolates arbitrarily and confidently. The uncertainty is the posterior variance in the case of vDUQ, and the variance across the ensemble element predictions in Deep Ensembles.

normalization enforces smoothness, while the residual connections enforce sensitivity of the feature represenation to changes in the input, obtaining a similar effect as the gradient penalty of DUQ. We use an inter-domain inducing point variational approximation of the GP predictive distribution (Lázaro-Gredilla & Figueiras-Vidal, 2009; Hensman et al., 2015), which places inducing points in feature space leading to needing fewer inducing points than previous work (Wilson et al., 2016a). These two techniques combined speed up inference in the GP model and decouple it from the dataset size. We release our code[1] and hope that it will become a drop in alternative for softmax models with improved uncertainty.

In Figure 1, we show how vDUQ and Deep Ensembles (Lakshminarayanan et al., 2017), the current state of the art for uncertainty quantification (Ovadia et al., 2019), perform on simple 1D regression. This task is particularly hard for deep networks as shown in Foong et al. (2019). vDUQ shows the desired behavior of reverting back to the prior away from the data, while the Deep Ensemble extrapolates arbitrarily and confidently. In between the two sinusoids, the Deep Ensemble is certain while vDUQ increases its uncertainty.

In summary, our contributions are as follows:

- We improve training a DKL model and for the first time match the accuracy and speed of training a deep network using regular softmax output on standard vision benchmarks.
- We demonstrate excellent uncertainty quantification in classsification which matches or exceeds the state of the art on CIFAR-10, including ensembling approaches.
- We show state of the art performance on causal inference for personalized medicine, an exciting real world application. This task requires calibrated uncertainty in regression to be able to defer treatment to an expert when uncertainty is high.

## 2 BACKGROUND

Gaussian Processes (GPs) provide an interpretable, explainable and principled way to make predictions, and can work well even with little training data due to their use of Bayesian inference. In contrast to deep neural networks, GPs have high uncertainty away from the training data and on noisy inputs.

There are however two main issues with the standard GP setup: poor performance on high dimensional inputs and inefficient computational scaling with large datasets. The poor performance on high dimensional inputs is due to the fact that most standard shift-invariant kernels are based on

---

[1]Available at: `anonymized-for-review`

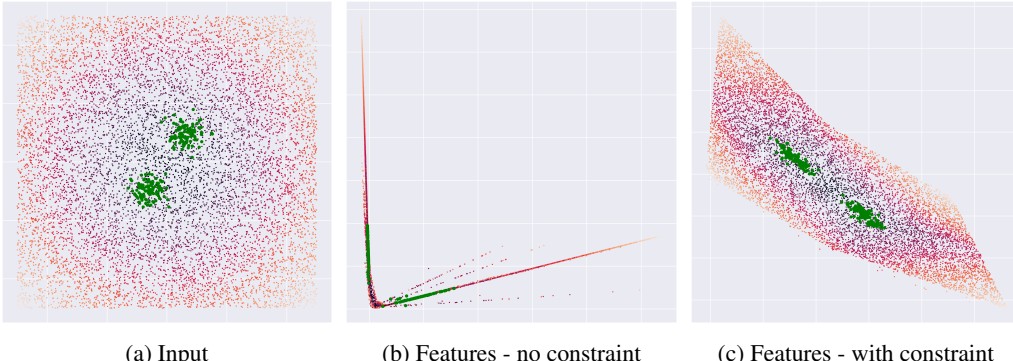

(a) Input        (b) Features - no constraint        (c) Features - with constraint

Figure 2: On the left we visualize a 2D classification task where the classes are two Gaussian blobs (drawn in green), and a large grid of unrelated points (colored according to their log-probability under the data generating distribution). In the center we see the features as computed by an unconstrained model. On the right the features computed by a model with residual connections and spectral normalization. As we can see, the objective for the unconstrained model introduces a large amount of distortion of the space, largely collapsing the input to a single line, making it almost impossible to use distance-sensitive measures on these features. In contrast, a constrained mapping still makes the classes more separable, while maintaining the relative distances of the other points.

the Euclidean distance, which is a poor metric for highly structured data like images (Theis et al., 2016). While kernels exist that better address this (Van der Wilk et al., 2017; Jacot et al., 2018), these are more computationally expensive and typically still underperform standard convolutional neural networks.

Deep Kernel Learning (Calandra et al., 2016; Hinton & Salakhutdinov, 2008; Wilson et al., 2016b) is a way to combine the expressiveness of deep neural networks with the attractive properties of Gaussian Processes. The core idea is to use a deep neural network inside the kernel of a GP, $k(\boldsymbol{x}_i, \boldsymbol{x}_j) \rightarrow k(f_\theta(\boldsymbol{x}_i), f_\theta(\boldsymbol{x}_j))$, where $f_\theta(\cdot)$ is a deep neural network, such as a Wide ResNet (Zagoruyko & Komodakis, 2016) up to the last linear layer, parametrized by $\theta$. The kernel $k(\cdot, \cdot)$ can be any of the standard kernels, such as the RBF or Matérn kernel. With the deep network it becomes possible to train the GP on datasets that contain high dimensional points, such as images. As the deep network is unconstrained, uncertainty away from the data collapses (Bradshaw et al., 2017), and the model can be arbitrarily confident while no data has been observed in that region during training. This has been a major drawback to using DKL in practice. In Section 3 we provide a detailed discussion of this effect and ways to mitigate it.

While DKL is a potentially powerful solution to overcome the problem of inexpressive kernels, it does not address the other scaling problem of Gaussian Processes: large datasets. The poor computational scaling comes from the fact that making predictions with an exact GP requires solving a linear system the size of the training data, namely computing $K(X, X)^{-1}K(X, x^*)$ where $K(X, X)$ is the kernel matrix evaluated on the training dataset $X$ and $K(X, x^*)$ the kernel between the training dataset and a test point $x^*$. This is a significant computational problem in its own right. A powerful and principled alternative to exact GP inference is inducing point GP approximate inference, where only a small set of inducing points in the input space $U$ are used to represent the entire training set, reducing the linear system to be solved to $K(u, u)^{-1}K(u, x^*)$. The new linear system is $m$ by $m$, where $m$ is the number of inducing points, rather than $N$ by $N$. Finding the optimal set of points, which are not necessarily part of our training set, can be done by treating them as variational parameters and maximizing a lower bound on the marginal log-likelihood (Titsias, 2009). We follow the variational inducing point formulation of SVGPC (Hensman et al., 2015) as implemented in GPyTorch (Gardner et al., 2018).

## 3   METHOD

vDUQ is an instance of Deep Kernel Learning, combining a constrained deep neural network with an inducing point GP. The model is learned end-to-end using variational inference by means of gradient

---

**Algorithm 1** Algorithm for training vDUQ

---

**Initialization:**
- Residual NN $f_\theta : x \to \mathbb{R}^d$ with feature space dimensionality d and parameters $\theta$ initialized using He et al. (2015).
- Approximate GP with variational parameters $\phi$ and number of inducing points $m$.

Using a random subset of $k$ points of our training data, $X^{\text{init}} \subset X$, compute:
**Initial inducing points:** K-means on $f(X^{\text{init}})$ with $K = m$. Use found centroids as initial inducing point locations in GP.
**Initial length scale:** $l = \frac{1}{\binom{k}{2}} \sum_{i=0}^{k} \sum_{j=i+1}^{k} |f(X_i^{\text{init}}) - f(X_j^{\text{init}})|_2$.

1: **for** minibatch $B_x = \{x_1, ..., x_b\}$ from $X$ and $B_y = \{y_1, ..., y_b\}$ from $Y$ **do**
2:    $\theta' \leftarrow$ spectral_normalization$(\theta)$
3:    $\psi \leftarrow f_{\theta'}(B_x)$                     ▷ Compute feature representation
4:    $p(Y'|B_x) \leftarrow \text{GP}_\phi(\psi)$       ▷ Compute posterior over labels, implemented in GPyTorch
5:    $L \leftarrow \text{ELBO}_\phi(p(Y'|B_x), B_y)$        ▷ Compute loss, implemented in GPyTorch
6:    $\phi, \theta \leftarrow \phi, \theta + \eta * \nabla_{\phi,\theta} L$            ▷ $\eta$ the learning rate, alternatively use ADAM
7: **end for**

---

descent. In this section we discuss how vDUQ overcomes previous problems with collapsing uncertainty in DKL (Bradshaw et al., 2017) and explain how to learn the model with no pre-training, few inducing points, and a standard minibatch size, avoiding some shortcomings of Wilson et al. (2016a).

Without restrictions, the deep network inside the kernel is free to map input points that are far away from the training distribution to a feature representation that resembles those of data points in the training distribution. This behavior is also exhibited in standard neural networks (Smith & Gal, 2018) and sometimes referred to as *feature collapse*. We visualize this in Figure 2, where we map a 2D input space into feature space. With the feature space of the unconstrained model it is impossible to recover uncertainty: many points are collapsed on top of each other.

In DUQ (van Amersfoort et al., 2020), the authors show that it is possible to reduce *feature collapse* by enforcing two constraints on the model: sensitivity and smoothness. Sensitivity implies that when the input changes the feature representation also changes: this means the model cannot simply collapse feature representations arbitrarily. Smoothness means small changes in the input cannot cause massive shifts in the output. This appears to help optimization, and ensures the feature space accords with the implicit assumptions that for example RBF kernels make about the data. We discuss a connection between these two requirements, sensitivity and smoothness, and bi-Lipschitz functions in Section 6.

There is a number of methods proposed in the literature that attempt to satisfy these constraints and each comes with different trade-offs:

- **Two-sided gradient penalty:** In DUQ (van Amersfoort et al., 2020), the properties are achieved by regularizing using a two-sided gradient penalty, that penalizes the squared distance of the gradient from a fixed value at every input point. This approach is easy to implement, but is not guaranteed to work, and in practice both the stability of training and its effectiveness as a regularizer can be fairly sensitive to the weighting of this penalty.

- **Direct spectral normalization and residual connections:** spectral normalization (Miyato et al., 2018; Gouk et al., 2018) on the weights leads to smoothness and it is possible to combine this with an architecture that contains residual connections for the sensitivity constraint (Liu et al., 2020). This method is faster than the gradient penalty, and in practice a more effective way of mitigating feature collapse.

- **Reversible model:** A reversible model is constructed by using reversible layers and avoiding any down scaling operations (Jacobsen et al., 2018; Behrmann et al., 2019). This approach can guarantee that the overall function is bi-Lipschitz, but the resulting model consumes considerably more memory and can be difficult to train.

In this work, we use direct spectral normalization and residual connections, as we find it to be more stable than a direct gradient penalty and significantly more computationally efficient than reversible models. In Figure 2, we show that a constrained model is unable to collapse points on top of each other in feature space, enabling the GP to correctly quantify uncertainty. Using a stationary kernel, the model then reverts back to the prior away from the training data just like a standard GP.

The regularized feature extractor allows us to offload computational complexity from the GP onto the deep model without sacrificing uncertainty. An expressive deep model is able to find a feature representation which generalizes across all intra class variation, and thus cluster inputs of the same class in feature space. We define the inducing points in feature space instead of input space (also known as an interdomain approximation) (Lázaro-Gredilla & Figueiras-Vidal, 2009; Hensman et al., 2017), which takes advantage of this clustering to reduce the number of inducing points required. In practice, we find that only very few inducing points, the number of classes in the case of classification, are necessary to obtain a well performing GP and we found similarly low numbers (in the 10 to 100 range) work well for regression, which means that solving the linear system is fast and the GP has minimal overhead compared to a softmax output. The fact that we can use few inducing points is not true in general for DKL (e.g. Wilson et al. (2016a) use several hundred points in input space), but requires the extra restrictions we have placed on our feature extractor to avoid the pathological feature collapse. We also find we can train with standard minibatch sizes of 128, while previous work used 5,000 on CIFAR-10 to combat gradient variance (Wilson et al., 2016a). It is important to note that in DKL and also in this paper we are not being Bayesian about our neural network parameters $\theta$, so we only need a single pass through the feature extractor both when training and during inference.

## 3.1 SPECTRAL NORMALIZATION IN BATCH NORMALIZATION LAYERS

When applying spectral normalization to the parameters of a deep model, it is important to note that batch normalization, a crucial component of training deep ResNets, has a non-trivial Lipschitz constant. In particular, since batch normalization transforms the input, using scale and shift parameters $\gamma$ and $\beta$, according to

$$x_{out} = \text{diag}\left(\frac{\gamma}{\sqrt{\text{Var}(x)}}\right)(x - \mathbb{E}[x]) + \beta, \tag{1}$$

it has a Lipschitz constant of $\max_i |\frac{\gamma_i}{\sqrt{\text{Var}(x)_i}}|$ (Gouk et al., 2018). Using the above equation, we can extend spectral normalization to batch normalization by dividing the weight $\gamma$ of the batch normalization by the (scaled) Lipschitz constant. In practice, we find empirically that typical batch normalization layers in trained ResNets have a relatively high Lipschitz constant, up to around 12, with 95% having a Lipschitz greater than one (see also Figure 5 in the Appendix). This is counter to the claim of Liu et al. (2020) that batch normalization reduces the Lipschitz value of the network. Following these claims, Liu et al. (2020) only constrain the convolutional layers, which as we demonstrate empirically results in models which are less sensitive to changes in the input, violating the constraint introduced in DUQ. Unless otherwise noted, we apply spectral normalization to both the convolutional and the batch normalization layers. For convolutional layers with $1 \times 1$ filters we use exact normalization, while for large filters we use an approximation, implemented originally by Behrmann et al. (2019).

## 4 RELATED WORK

Currently, the simplest and most effective way to obtain uncertainty in deep learning classification is using ensembling (Lakshminarayanan et al., 2017). Despite its simplicity it has shown to be remarkably effective (Ovadia et al., 2019), outperforming alternatives such as MC Dropout (Gal & Ghahramani, 2016) and mean-field variational inference Bayesian Neural Networks (Farquhar et al., 2019) at the expense of having to train and evaluate multiple models. An important issue with Deep Ensembles is enforcing diversity; while on large and complicated datasets it is sufficient to use a different initialization and data order, this is not sufficient on easier or smaller datasets, as we highlight in Figure 1.

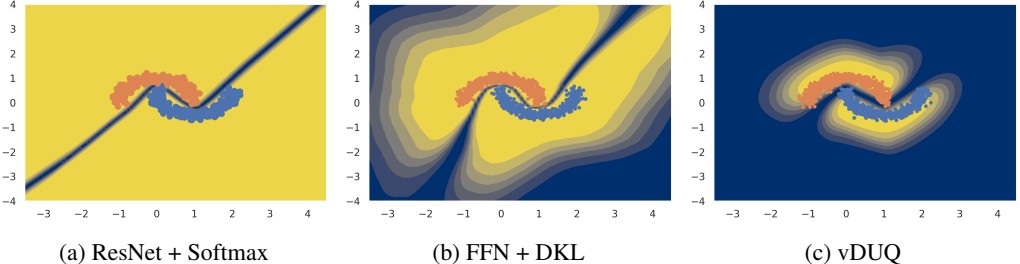

|  (a) ResNet + Softmax | (b) FFN + DKL | (c) vDUQ |

Figure 3: We show uncertainty results on the two moons classification dataset. Yellow indicates high confidence, while blue indicates uncertainty. In Figure 3a, a simple feed forward resnet with a softmax output is certain everywhere except on the decision boundary. In Figure 3b, we see that a simple feed forward model in combination with DKL has large areas of certainty, even away from the training data. In Figure 3c, we show vDUQ with a residual network as feature extractor in combination with spectral normalization, showing close to ideal uncertainty on this dataset.

van Amersfoort et al. (2020) demonstrate empirically with DUQ that it is possible to quantify uncertainty in deep learning using only a single deterministic model. SNGP (Liu et al., 2020) extends on DUQ's ideas of a distance aware prediction function and spectral regularization, and implements them by using a deep neural network with a random fourier features (RFF) approximate GP and direct spectral normalization on the convolutional weights (but not on the weights of batch normalization). While they obtain impressive results, it is important to note that the RFF approximation makes the GP parametric, meaning there are now a fixed number of basis functions used in the prediction which can lead to unreliable uncertainty. In contrast, our use of variational Bayes in the form of SVGPC conserves the non-parametric property and uses infinitely many basis functions; it is an approximation to the true GP posterior, whereas Liu et al. (2020)'s random Fourier expansion leads to a *different* model which converges in expectation to the true model as the number of random Fourier features tends to infinity. This is discussed in Section 6 of Lázaro-Gredilla et al. (2010) which comment that the RFF approximation is at risk of overfitting, while the variational inducing point approximation is safeguarded from it as they are parameters of the variational approximation (Titsias, 2009). Furthermore, the RFF approximation is only tractable for the RBF kernel and significantly more expensive for other kernels, restricting the flexibility of the model. Finally, although in principle the method of Liu et al. (2020) could be adapted to regression in a straightforward way, they do not investigate this. Our experiments demonstrate the applicability of this kind of model both on standard classification benchmarks and uncertainty sensitive regression tasks, such as causal inference.

## 5 EXPERIMENTS

### 5.1 REGRESSION AND IN-BETWEEN UNCERTAINTY

In Figure 1, we show results on a 1D dataset created from sinusoids. For vDUQ, we use spectral normalization, a Matérn kernel with $\nu = \frac{1}{2}$ and 20 inducing points. This kernel is known to lead to less smooth learned functions, which leads to quickly increasing the uncertainty when there is no data. Full experimental details are provided in Appendix A

Deep Ensemble extrapolates incorrectly as $x$ becomes more negative, at $x = -15$ the prediction is far away from what is seen in the training data, however the model still has low uncertainty. Meanwhile vDUQ reverts to the prior directly at the edge of the training data domain, which is the ideal behavior in this regression setting. In Deep Ensembles (Lakshminarayanan et al., 2017), it was suggested to let the model predict both the mean and the standard deviation. This standard deviation only measures the *aleatoric* uncertainty (see also Chua et al. (2018) for a discussion), which is the uncertainty arising from noise inherent in the data. In this dataset there is only minimal noise, and therefore the only way to assess the uncertainty is to compute the variance of the mean predictions. In contrast, vDUQ's posterior variance is directly interpretable as the predictive uncertainty (also known as the total uncertainty). In the Appendix, Figure 4, we visualize sampling from the posterior, which highlights the non-smoothness of the kernel choice.

Table 1: Results on CIFAR-10 and distinguishing the test set of CIFAR-10 and SVHN by uncertainty (AUROC). All results are the average and standard deviation of 5 runs.

| Method | Accuracy (%) | AUROC |
|---|---|---|
| WRN (Zagoruyko & Komodakis, 2016) | **96.2±0.03** | 0.932±0.018 |
| Ensemble of 5 WRN (Lakshminarayanan et al., 2017) | **96.6±0.06** | **0.967±0.011** |
| SNGP (Liu et al., 2020) [2] | 96.0±0.09 | 0.940±0.013 |
| DUQ (van Amersfoort et al., 2020) | 94.6±0.16 | 0.940±0.007 |
| **vDUQ** (no Spectral Normalization) | **96.2±0.11** | 0.937±0.014 |
| **vDUQ** | 95.6±0.09 | **0.958±0.012** |

## 5.2 CLASSIFICATION ON TWO MOONS

We show results on Two Moons dataset (Pedregosa et al., 2011) for three different models: a standard softmax model, vDUQ, and a variation where the spectral normalized ResNet is replaced by a fully connected model (similar to Bradshaw et al. (2017)). For experimental details see Appendix A.

On this simple dataset, a multi layer neural network easily achieves 100% accuracy with a single, non-linear, decision boundary going between the two moons. However these models exhibit complete certainty everywhere in the input domain as shown in Figure 3a, with yellow the certain regions, and blue uncertain. The uncertainty is computed using the entropy of the class prediction: we model the problem as a two class classification problem. vDUQ (in Figure 3c) on the other hand quantifies uncertainty exactly as one would expect for the two moons dataset: certain on the training data, uncertain away from it and in between the two half moons. Figure 3b highlights the importance of our contribution. When using DKL with a deep network without the right constraints, the model generalizes beyond the training distribution and is certain even away from the training data.

## 5.3 CIFAR-10 AND SVHN

In this section we look at training a large model, the Wide Residual Network, on CIFAR-10 (Krizhevsky et al., 2009). For vDUQ, we train the large ResNet end-to-end with the GP, this is in contrast with prior work that used a pre-trained model (Bradshaw et al., 2017), which limits the ability to apply the method to new domains. We follow the experimental setup of Zagoruyko & Komodakis (2016), and use a 28 layer model with BasicBlocks and dropout. Interestingly, we can follow hyper parameter suggestions (such as dropout rate, learning rate, and optimizer) directly when training vDUQ, and no special care is necessary for the variational parameters. We remove the final linear layer of the model and the resulting 640 dimensional feature vector is directly used in the GP. We train using only *10 inducing points*, which are shared among the independent output GPs (although $q(u)$ is different) and use Lipschitz factor 3. This means the model is fast, and going through one epoch is just 3% slower than using a softmax output. Further experimental details are discussed in Appendix A. We provide an ablation study of the number of inducing points in Appendix C.

Results are shown in Table 1. vDUQ without any spectral normalization matches the accuracy of the WRN, but the uncertainty is only slightly better than a softmax model. With spectral normalization the accuracy drops slightly, as expected, but the uncertainty improves significantly. Note that SV-DKL (Wilson et al., 2016a), our most direct baseline, obtains just 91% accuracy on CIFAR-10 using a ResNet-20 (for which the softmax accuracy is 94%, see also van Amersfoort et al. (2020)). Meanwhile, convolutional GPs (Van der Wilk et al., 2017) obtain 64.6% accuracy while using 1,000 inducing points. We perform an ablation where we train vDUQ with spectral normalization only on the convolutions, similar to (Liu et al., 2020), and we obtain an AUROC of $0.93 \pm 0.003$ and accuracy matching the model without spectral normalization. This provides further evidence that the batch normalization layers undo the effect, and need to be properly normalized.

---

[2]Obtained using the author's open source implementation, which is available at `https://github.com/google/uncertainty-baselines`

Table 2: Comparing the performance of several causal effect inference methods. Best results in bold and note that DKLITE* results are obtained using their open source implementation (see A.3).

| $\sqrt{\epsilon_{\text{PEHE}}}$ Method / Policy | CEMNIST (50% deferred) random | uncertainty | IHDP Cov. (50% deferred) random | uncertainty | IHDP (10% deferred) random | uncertainty |
|---|---|---|---|---|---|---|
| BART | 2.1±.02 | 2.0±.03 | 2.6±.2 | 1.8±.2 | 1.90±.20 | 1.60±.10 |
| BT-Learner | .27±.00 | .04±.01 | 2.3±.2 | 1.3±.1 | 0.95±.03 | 0.69±.01 |
| BTARNet | .18±.01 | **.00±.00** | 2.2±.3 | **1.2±.1** | 1.10±.03 | 0.76±.03 |
| BCFR-MMD | .32±.01 | .13±.02 | 2.5±.2 | 1.7±.2 | 1.30±.06 | 0.91±.03 |
| BDragonnet | .22±.01 | .02±.01 | 2.4±.3 | 1.3±.2 | 1.50±.05 | 1.05±.02 |
| BCEVAE | .30±.01 | .04±.01 | 2.5±.2 | 1.7±.1 | 1.80±.06 | 1.47±.08 |
| DKLITE* | | | 2.6±.7 | 1.8±.5 | 1.74±.53 | 1.34±.41 |
| **vDUQ** | **.16±.02** | .01±.00 | **1.82±.35** | **1.23±.16** | **0.90±.18** | **0.53±.09** |

## 5.4 CAUSAL INFERENCE

Personalized healthcare is an exciting application of machine learning, where the efficacy of a treatment is predicted based on characteristics of the individual using a model trained on previously treated patients. An individual's response to treatment can only be known if they are a member of a group represented in the data and if there is prescription diversity in the group: treatment and no treatment. Jesson et al. (2020) show that measures of uncertainty can identify when personalized causal-effect inference fails due to such factors. Uncertainty can then define policies for deferring treatment recommendations to an expert when there is insufficient knowledge about a person. The uncertainty estimates must be correct, or else individuals would receive treatment recommendations even if their response to treatment is not known, which can result in undue stress, financial burdens, or worse. In this section we evaluate vDUQ on the task of inferring the personalized causal-effects of a binary treatment $t$ on an individual $x$ via the conditional average treatment effect: $\text{CATE}(x) := \mathbb{E}[Y_{t=1}(x) - Y_{t=0}(x)]$ (Abrevaya et al., 2015). Intuitively, an individual would be recommended treatment if $\text{CATE}(x) > 0$. We show that vDUQ yields uncertainty estimates that lead to safer and more data-efficient policies for withholding treatment recommendations than alternative methods *significantly improving on previous state-of-the-art results*.

We use the semi-synthetic IHDP (Hill, 2011), IHDP Cov. and CEMNIST datasets to assess the uncertainty estimates of vDUQ on this causal inference task following the experimental setup in (Jesson et al., 2020). IHDP and IHDP Cov. are regression datasets with only $\sim$750 data points, while CEMNIST is a binary classification dataset with 10k points. Treatment-effect recommendations are deferred to an expert either at *random*, or based on the *uncertainty* in the CATE estimate. Recommendations are deferred for **10%** of the cases in the IHDP dataset, and **50%** of the IHDP Cov. and CEMNIST datasets. We report the root expected Precision in Estimation of Heterogeneous Effect (Hill, 2011) ($\sqrt{\epsilon_{\text{PEHE}}}$) to assess the error on the remaining recommendations (lower is better). Table 2 summarizes our results and shows that vDUQ has improved performance and uncertainty estimates better suited to rejection policies than other uncertainty aware methods. In particular, we improve on DKLITE (Zhang et al., 2020), the baseline related to our method: an alternative deep kernel learning method designed specifically for CATE inference.

## 6 LIMITATIONS

Even though demonstrated to work well empirically above, a theoretical justification of the spectral normalization in combination with residual connection is still lacking. Liu et al. (2020) propose an explanation based on the fact that spectral normalization, in combination with residual architectures, induces a bi-Lipschitz constraint, which is defined as:

$$\frac{1}{K} d_X(x_1, x_2) \leq d_Y\left(f_\theta(x_1), f_\theta(x_2)\right) \leq K d_X(x_1, x_2). \tag{2}$$

From that constraint it follows that if we have two metric spaces $X$ and $Y$, with a continuous function $f : X \mapsto Y$ between them which is bijective, and $f$ is bi-Lipschitz, then metric properties of $X$ like boundedness and completeness have to be preserved by $Y$, and distances in $X$ can be changed only by a bounded factor in $Y$. In this sense, a bi-Lipschitz map between $X$ and $Y$ can be considered to

be "approximately" distance preserving. Liu et al. (2020) argue that this means that applying this penalty to a Wide ResNet causes it to be a distance preserving mapping. However, the proof they provide is only valid for spectral regularisation with coefficients less than 1, whereas empirically we find that exceeding 1 has similar effectiveness in terms of uncertainty and is easier to optimize. This suggests that the proof does not capture the underlying mechanism of the spectral normalization. In addition, this argument is not applicable to models that do any form of downsampling as it is straightforward to demonstrate the following observation (which we do not claim to be novel):

**Proposition 1.** *Let $f$ be a function from $\mathbb{R}^n$ to $\mathbb{R}^m$, with $m < n$. Then $f$ is not bi-Lipschitz.*

We provide a proof in Appendix B; it suffices to show that a bi-Lipschitz function is necessarily a homeomorphism, and so cannot exist between spaces of different dimensionality.

We thus consider explanations of the effectiveness of spectral normalization and residual connections based on the assumption that the bi-Lipschitz condition *holds* insufficient if the network uses dimensionality reduction, which most feature extraction architectures do. Some progress towards a well performing bijective deep model was made in Behrmann et al. (2019), however the iResNet is significantly slower and more computationally expensive than the WRN, and we were unable to achieve the performance we report here without using dimension-reducing layers. Despite these open theoretical questions, we have so far not found a practical example where the uncertainty quality was poor, and our empirical results call for further theoretical study.

## 7 CONCLUSION

We present vDUQ, a practical probabilistic method which allows for using deep architectures and we established its efficacy in 2D, on a large image dataset and in causal inference for personalized medicine, obtaining or matching state of the art results in each. Good uncertainty and fast inference in both classification and regression make vDUQ a compelling plug-in option for many applications, such as exploration in reinforcement learning, active learning, and many real world tasks. Exciting directions for future research include a rigorous theory of why spectral normalization works despite downsampling, or alternatively a computationally efficient way to enforce a bi-Lipschitz constraint.

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

## A    EXPERIMENTAL DETAILS

### A.1    2D EXPERIMENTS

We perform our 2D experiments using a simple feed forward ResNet. The first linear layer maps from the input to the feature representation and does not have an activation function. From there on, the model is a ResNet, $x' = x + f(x)$, with $f(\cdot)$ a combination of a linear mapping and a relu activation function. The linear mapping has optional Spectral Normalization for which we use the implementation of Behrmann et al. (2019). We use the SGD optimizer for regression and Adam for the two moons with learning rate 0.01, and we use 4 layers with 128 features. For the toy regression, we use a max Lipschitz constant 0.95, one power iteration and 20 inducing points. For the two moons, we set the noise level to 0.1 and use five power iterations, a Lipschitz constant of 0.65 and four inducing points. For the Deep Ensemble in Figure 1 we train 10 separate models, using a different initialization and data order for each and train to minimize the squared error.

### A.2    WIDE RESNET EXPERIMENTS

For the WRN, we follow the experimental setup and implementation of Zagoruyko & Komodakis (2016). This means that for CIFAR-10, we use depth 28 with widen factor 10 and BasicBlocks with dropout.

We train for the prescribed 200 epochs with batch size 128, starting with learning rate 0.1 and dropping with a factor of 0.2 at epoch 60, 120 and 160. We use the full training set and take the model at epoch 200 (no early stopping).

For Spectral Normalization, we again use the implementation of Behrmann et al. (2019), in particular the convolutional version and also constrain batch normalization as described in (Gouk et al., 2018). We use 1 power iteration and use the lowest Lipschitz constant that still allows for good accuracy, which we found to be around 3 in practice. We increase the momentum of batch normalization to 0.99 (from 0.9 default) to reduce the variance of the running average estimator of the empirical feature variance, which can be a source of instability (Gouk et al., 2018).

### A.3    CAUSAL EXPERIMENTS

Following Shalit et al. (2017), we use 63/27/10 train/validation/test splits and report the PEHE evaluated on the test set over 1000, and 20 trials for the IHDP and CEMNIST datasets, respectively. For each trial, we train for a maximum of 750 epochs and evaluate the model with the lowest ELBO evaluated over the validation set. We employ Adam optimization with a learning rate of 0.001 and batch size of 100.

The feature extractor uses a feed forward ResNet architecture with 3, 200 unit hidden layers, and ELU activations. Dropout is applied after each activation at a rate of 0.1. The feature extractor takes the individual $x$ and treatment $t$ as input. For the CEMNIST experiment, a depth 3 CNN Resnet is used to extract features from the image, which is then passed as an input to the above architecture. Spectral normalization is used on all layers of the feature extractor. We use a Matérn kernel with $\nu = \frac{1}{2}$, 200 inducing points, and a smoothed box prior on the lengthscales with range $(\exp(-1), \exp(1))$.

For the DKLITE experiments we use the open source code from the authors available at `https://bitbucket.org/mvdschaar/mlforhealthlabpub/src/master/alg/dklite/`. We write a custom loop over the IHDP dataset to follow the above protocol. For DKLITE on CEMNIST, in initial experimentation we found that the method did not adapt well to images, so we omit the comparison from the table. We make this adaptation available at `anonymized-for-review`.

### A.4    INDUCING POINT GP

We initialize the inducing points by doing k-means on the feature representation of 1,000 points. We compute the initial length scale by taking the pairwise euclidean distance between feature representation of the 1,000 points. We whiten the inducing points before training, as suggested in Matthews

(2017). The kernel parameters, such as the length scale and output scale, are different per GP. We implement it using GPyTorch (Gardner et al., 2018) and use their default values if not otherwise specified.

## B  PROOF OF PROPOSITION 1

This proof makes no claim to novelty, but is provided for completeness.

*Proof of proposition 1.* We can show that this follows from the fact that the metric spaces $\mathbb{R}^m$ and $\mathbb{R}^n$ are not homeomorphic (topologically equivalent) to one another (Brouwer, 1911). While this statement is extremely intuitive, proving it is surprisingly technical, so we will take it as given.

First, we prove the following lemma.

**Lemma 1.** *Let $f : X \mapsto Y$ be a bi-Lipschitz and onto function, so $\frac{1}{L}||x_1 - x_2||_X \leq ||f(x_1) - f(x_2)||_Y \leq L||x_1 - x_2||_X$, and $Y$ is the image of $X$ under $f$. Then $f$ is a homeomorphism between $X$ and $Y$, and so $X$ and $Y$ are homeomorphic.*

*Proof.* Recall that a function $f$ is a homeomorphism if $f$ is bijective, $f$ is continuous, and $f^{-1}$ is also continuous. We will address these in turn. To see that $f$ is bijective, note that $f$ is injective iff $\forall x_1, x_2 \in X$, we have $x_1 \neq x_2 \implies f(x_1) \neq f(x_2)$. But this follows directly from the lower Lipschitz property of $f$, since if $x_1 \neq x_2$, then $||x_1 - x_2||_X > 0$, so $||f(x_1) - f(x_2)||_Y > 0$, from which it follows that $f(x_1) \neq f(x_2)$. Since $f$ is injective (one-to-one) and onto, it is a bijection. Since any function which is Lipschitz continuous is also continuous, the fact that $f$ is continuous is given. Since $f$ is bijective, the inverse function $f^{-1}$ exists, and we need to show that it is continuous. We have, from the bi-Lipschitzness of $f$, that $\frac{1}{L}||f^{-1}(f(x_1)) - f^{-1}(f(x_1))||_X \leq ||f(x_1) - f(x_2)||_Y$, which implies that the inverse function is also Lipschitz, and hence also continuous. So $f$ is a homeomorphism, and we are done. □

We have therefore established that if a bi-Lipschitz (and onto) function exists between two spaces $X$ and $Y$, then $X$ and $Y$ are homeomorphic. This provides a proof of our claim by contradiction; if a function $f : \mathbb{R}^n \mapsto \mathbb{R}^m$ existed and was bi-Lipschitz, then we would have shown that $\mathbb{R}^n$ and $\mathbb{R}^m$ were homeomorphic, since any bi-Lipschitz function is a homeomorphism. But $\mathbb{R}^n$ and $\mathbb{R}^m$ are not homeomorphic unless $m = n$, so no such function can exist. □

## C  ABLATION OF NUMBER OF INDUCING POINTS

In Table 3, we show the results of increasing the number of inducing points. While more inducing points generally helps, the differences are minimal.

Table 3: Accuracy on CIFAR-10 for increasing number of inducing points ($k$). The results are an average of 5 runs and reported with standard deviation.

| $k$ | Accuracy (%) |
|---|---|
| 10 | 96.12±0.13 |
| 20 | 96.07±0.12 |
| 50 | 96.17±0.14 |
| 100 | 96.25±0.08 |

## D  SAMPLING FROM THE JOINT POSTERIOR

In Figure 4, we show samples from the joint predictive posterior. This type of sampling is not possible using standard deep learning models, because there is no covariance modelled between the batch elements. Possible applications of this include batch active learning, where we want to estimate a joint posterior (Kirsch et al., 2019), but also for deep exploration in RL where we can effectively sample a full policy (Osband et al., 2016).

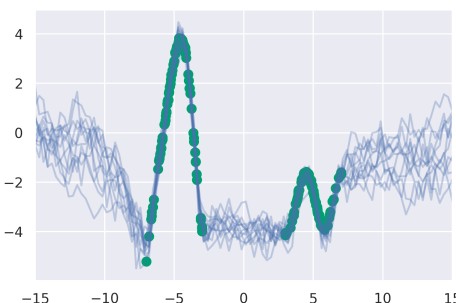 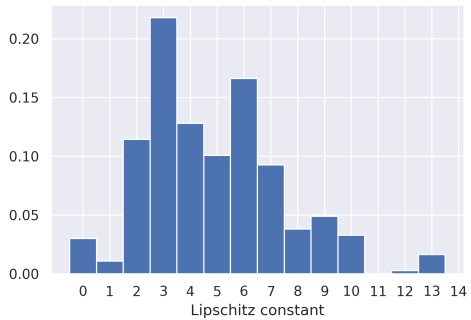

Figure 4: The same setup as in 1b, but instead of visualizing the uncertainty directly, we visualize samples from the joint posterior. As expected, the functions become significantly less smooth away from the data as expected using the Matérn $\nu = \frac{1}{2}$ kernel.

Figure 5: A density of the Lipschitz values in batch normalization layers, averaged across 15 WRN models that were trained with Softmax output and without Spectral Normalization (exactly following Zagoruyko & Komodakis (2016)). We see that many of the constants are significantly above 1, invalidating the claim that Batch Normalization reduces the Lipschitz constant of the network.

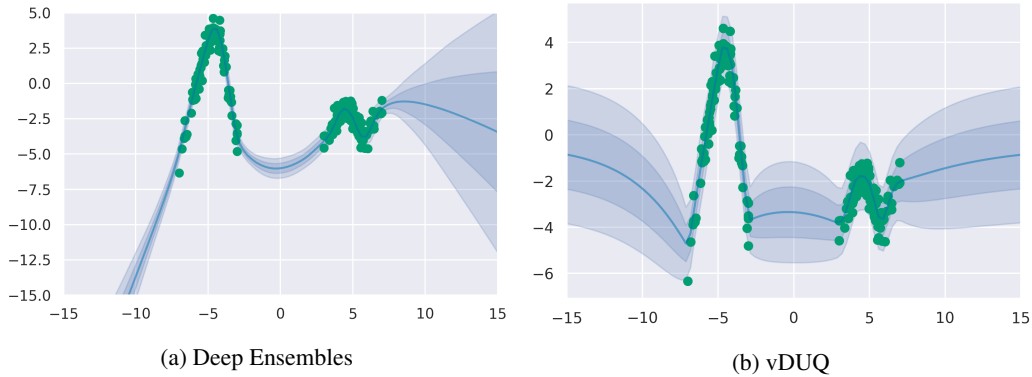

(a) Deep Ensembles

(b) vDUQ

Figure 6: The same setup as in Figure 1b, but with noise sampled from $N(0, 0.5)$ instead of $N(0, 0.05)$.

