# OpenReview forum: "Variational Deterministic Uncertainty Quantification"
_ICLR.cc/2021/Conference — Reject_

### Official Review · AnonReviewer2 · 2020-10-28
**Interesting idea that needs polishing in terms of presentation and empirical evaluation**

**Rating:** 5
**Confidence:** 4

**Review:**

Variational deterministic uncertainty quantification

Summary:
The paper proposes a method for out-of-distribution detection by combining deep kernel learning and Gaussian processes. Using neural networks as a kernel for the GP as well as inducing point approximation alleviates the scalability issues of GP. The idea itself has merits, however, the presentation and experiments are not convincing.

Strengths: The idea of using deep kernels within GP is a good solution that allows benefiting from both the expressiveness of the kernels and uncertainty estimates for GP. Additionally, using the uncertainty estimates for causal inference is a nice application.

Weaknesses: Although the approach is interesting it needs to be further developed and evaluated in multiple setups. I find it limiting that it relies on the residual connection, making it unsuitable for other NN architectures, which means it will apply to only a limited number of tasks.


The presentation of the method should be better structured. I appreciate the background on deep kernels and how it helps to overcome the limits of GP, however, there is a lack of presentation of the method itself. A description, algorithmic listing or even an equation for the uncertainty score proposed is missing in the current version of the text.

In the introduction vUQD is presented as favorable wrt UQD due to its rigorous probabilistic interpretation, however, this was never further analyzed in the text. Also, seems that the method is concerned only with the epistemic uncertainty in the data? In general, the whole presentation of related work and positioning of this paper in the uncertainty literature is not clear. What source of uncertainty does the method address? There is much to be elaborated on this topic and I believe the discussion on this will significantly improve the paper.

The discussion on spectral-normalization and bi-Lipschitz in 3.1
Please clarify it or explain it better, in the current writing it is contradicting the proposed method:
“A complete, rigorous theory of why the spectral normalization as used in this and previous work is a useful regularization scheme is not remains an open question”

Experiments:

Toy examples:
Figure 1 - on regression, I do not find this example motivating, first, why choosing noiseless data? Second, why is the vUQD increasing in reasons where there is data (such as the peaks?) Why does it compare only to deep ensembles?
Figure 2 - Why choosing a toy example where a linear classifier works in the original space?

What is the sensitivity to the number of inducing points for the GP? An ablation study at least for the toy data sets can help.

Why were standard datasets such as MNIST and fashion MNIST not included?
The empirical evaluation should be extended with more baselines and datasets.


Minor:
The manuscript needs proofreading, language errors increase increasingly towards the conclusion.

------------- Update after reading authors response -------------

I thank the authors for their detailed responses, they have answered most of my concerns and I raise my score to 5. I am still not convinced about the method covering both the aleatoric and epistemic uncertainties, without any theoretical or intuitive justification, and without any discussion/clarification on that part. If indeed this is the case, then additional experiments should be included, for example for a regression task, the standard UCI datasets [1].

[1] Hernandez-Lobato, J M and Adams, R P. Probabilistic ´
backpropagation for scalable learning of bayesian neural networks. In ICML-15, 2015.

---

> ### Author Response · Authors · 2020-11-17
> **Response**
>
> We thank the reviewer for taking the time to review our work and for your comments.
>
> We are surprised to hear the reviewer finds residual connections to be limiting since many modern networks utilize such connections. The reviewer mentions that residual connections limit the applicable tasks to which this model can be applied - we are unsure why this would be the case, as the residual architecture can be widely adapted to different tasks - for example, to linear layers and convolutional layers of any dimensionality. Could the reviewer give more detail on their concerns about the applicability?
>
> We note that, as an alternative, a direct gradient penalty as used in DUQ (van Amersfoort et al., 2020) could be used with no restrictions on the architecture. We discuss this alternative and limitations in Section 3 of the paper.
>
> We have added an algorithm box describing the process of training vDUQ step by step. It reduces to a simple gradient descent on a particular loss, which contributes to the wide applicability of our suggested method. The uncertainty in a GP can be computed as the entropy of the softmax (in the classification case) or using the variance of a prediction (in the regression case). In the regression case this can be done analytically.
>
> vDUQ is able to measure both aleatoric and epistemic uncertainty, in similar fashion to Deep Ensembles and MC dropout. In DUQ, the uncertainty is defined as the distance to the closest centroid. This ignores the fact that a point can be close to two centroids, and therefore makes it difficult to quantify aleatoric uncertainty. We did not elaborate on this distinction for vDUQ since by design it covers all sorts of uncertainty.
>
> The discussion on spectral-normalization and bi-Lipschitz in 3.1 Please clarify it or explain it better, in the current writing it is contradicting the proposed method: “A complete, rigorous theory of why the spectral normalization as used in this and previous work is a useful regularization scheme is not remains an open question”
>
> On *discussion of spectral normalization*: the cited sentence is a typo, please strike out “is not” in the last part of the sentence. We have fixed this in the paper
>
> Could you give more detail on what you find unclear about this section - do you find it difficult to follow, or not understand the conclusions?
> We have moved the theoretical discussion of the bi-lipschitz constraint to a separate Limitations section (see Section 6 in the updated paper) as it is not directly related to the method. We instead include a description of the properties that need to hold for the neural network such that we avoid feature collapse and provide empirical proof that this happens.
>
> ~~~
> Toy examples: Figure 1 - on regression, I do not find this example motivating, first, why choosing noiseless data? Second, why is the vUQD increasing in reasons where there is data (such as the peaks?) Why does it compare only to deep ensembles? Figure 2 - Why choosing a toy example where a linear classifier works in the original space?
> ~~~
>
> We have added noise to the data and obtained the same result (see figure 6 in the appendix). We have updated figure 1 with a better fit vDUQ that matches the peaks behavior of Deep ensembles. We compare to Deep Ensembles because they are the current state of the art method in uncertainty quantification in deep learning.
>
> On *sensitivity to the number of inducing points*: the fact that we can train to match softmax accuracy (96.2% accuracy on CIFAR-10) with only 10 inducing points implies that we are very robust. We have added results for more inducing points are added as Appendix C. in the paper.
>
> Instead of MNIST, we emphasized the significantly more difficult CIFAR-10 experiment, and instead opted for an interesting application of regression uncertainty in Causal Inference. MNIST and FashionMNIST are useful for development, but we believe MNIST vs FashionMNIST is too easy to have much practical value and is not a reliable way to compare different methods. If the reviewer thinks there is value in this experiment, we would be glad to add it.

---

### Official Review · AnonReviewer1 · 2020-10-28
**Official Blind Review #1**

**Rating:** 5
**Confidence:** 4

**Review:**

---
##### Summary:

This paper proposes variational deterministic uncertainty quantification (vDUQ), which adopts the stochastic (sparse) variational deep kernel learning (DKL) method to enable uncertainty estimations for deep models. To avoid uncertainty collapse, the deep neural network in the GP kernel is regularized with spectral normalization, which ensures a bi-Lipschitz constraint. Experiments show that vDUQ is effective in uncertainty quantification tasks.

##### Reasons for score:

The idea is clear and the paper is easy to follow. However, my major concern is about the significance of the contribution and the experimental results. See my detailed comments below.

---
##### Pros:

The idea of obtaining uncertainty estimations with DKL is interesting. Moreover, by using sparse variational inference in DKL, the entire model is just like a DNN with an extra “layer of inducing points”, requiring only a few extra parameters and computational cost, which is also desired. Overall, the paper is well-written. The figures are instructive and helpful for understanding.

---
##### Concerns:

My main concern is about the significance of the contributions. Sparse variational inference methods for DKL was previously proposed in Wilson et al. 2016a. The main contribution of this paper seems to be the idea of introducing the spectral normalization regularization to stabilize the training of DKL and avoid uncertainty collapse. Although this is an interesting idea, I think the authors did not provide clear enough explanations and rigorous analyses.

In Page 3 the authors mentioned “without the spectral regularization on the deep model, the deep network is free to map data points that are far away from the training distribution to a feature representation that’s similar to in distribution data”. This perhaps explains how uncertainty collapse happens in out-of-distribution data to some extent, but is this the primary cause of undesired uncertainties?

Since the parameters of NN become the parameters of the GP prior (as mentioned in Section 2), optimizing the marginal likelihood or the ELBO w.r.t the variational parameters and the NN parameters is actually “fitting the prior to data”, which could also cause biased uncertainties [1]. Although the bi-Lipschitz property can intuitively alleviate the biases, it is not clearly explained how it works. It would be better to provide more theoretical analysis.

In Section 3.1, the authors raised a question about how informative the “distance preserving explanation” is about the motivation of using bi-Lipschitz regularization. However, a more informative explanation is not provided. Also, the last paragraph of Section 3.1 is misleading. The author mentioned “A complete, rigorous theory … is not remains an open question.” If it is addressed and has theoretical insights into the use of spectral normalization, the authors should add necessary references and explanations.

[1] Salimbeni, Hugh, and Marc Deisenroth. "Doubly stochastic variational inference for deep Gaussian processes." Advances in Neural Information Processing Systems. 2017.


Minors:
(1) In Table 1, the results of vDUQ and DUQ is outperformed by the ensemble method in terms of both accuracy and AUROC. This seems a different conclusion form the results in Amersfoort et al. 2020. It would be better to provide more discussion about it, which seems not to be expected.

(2) The authors claim the vDUQ can be trained in an end-to-end fashion in Section 3. However, since the inducing points are initialized with k-means algorithms that need to look at the training data, which I think is still a (lightweight) pre-training.

---

> ### Author Response · Authors · 2020-11-17
> **Response**
>
> We are glad that the reviewer found our paper clear and easy to follow, and that they think that our central idea is interesting and worth exploring.
>
> On *comparison with SV-DKL*, previous work on VI in DKL (Wilson et al., 2016) used inducing points in *input space* - we use an interdomain approximation (La ́zaro-Gredilla & Figueiras-Vidal, 2009) and place them in feature space. This dramatically reduces the number of inducing points and mini batch size necessary, from several hundred to tens, and from 5,000 to 128, respectively.
> Our changes make DKL much more straightforward to train. In combination with correcting DKL’s problem with feature collapse we obtain a model that is very effective in practice, obtaining state of the art uncertainty for a single model.
>
> ~~~
> In Page 3 the authors mentioned “without the spectral regularization on the deep model, the deep network is free to map data points that are far away from the training distribution to a feature representation that’s similar to in distribution data”. This perhaps explains how uncertainty collapse happens in out-of-distribution data to some extent, but is this the primary cause of undesired uncertainties?
> ~~~
>
> The reviewer makes an interesting point about whether uncertainty collapse is the primary cause of undesirable uncertainty behavior. As they point out elsewhere in their review, this is likely not the only cause of model overconfidence - for instance, it is still possible that our method could have miscalibrated uncertainty due to the ‘fitting the prior’ effect of learning a neural kernel, as they point out. However, this is an issue with any deep kernel learning method, so we do not believe this should take away from our paper significance. One does not need to prevent all undesirable properties of uncertainty in order to demonstrate material gains in down-stream tasks, as we show empirically. Our empirical results suggest that spectral regularization provides significant improvements in uncertainty quality, which would imply that this behavior is an important cause of miscalibrated uncertainty in practice as also demonstrated in previous works (van Amersfoort et al., 2020) and (Liu et al., 2020).
>
>  On *explanation of bi-Lipschitz property*: we have rewritten the methods section to highlight the two constraints we need to put on the deep model in order to have high quality uncertainty from DKL. These are:
> 1. Sensitivity, the feature representation changes if the input changes (avoiding feature collapse).
> 2. Smoothness, the feature representation does not change too much for small changes in the input.
> In previous work, DUQ (van Amersfoort et al., 2020) find empirically that spectral regularization (by means of a two-sided gradient penalty) achieves this. We describe this method and two alternatives, of which we use the direction spectral normalization version, of which the efficacy to obtain the above properties was shown in (Liu et al., 2020).
> We created a limitations section (section 6.) that addresses the theoretical discussion.
>
> On *last paragraph of Section 3.1*: this last sentence is a typo (strike out “is not”), which we apologize for and have fixed. We have moved some theoretical discussion to Section 6 Limitations.
> However, we are curious why the author feels that this is misleading - we point out that we do not have a watertight theoretical justification for our penalty, but empirically we demonstrate SOTA performance across a range of tasks. Does the reviewer feel that this is insufficient justification? We would appreciate a clarification of their criticism.
>
> Minor (1) - van Amersfoort et al, 2020, come to the same conclusion in the context of CIFAR-10 in table 3 of their paper (Deep Ensembles outperforms DUQ on AUROC)
> Minor (2): We perform the K-means with only 1000 training points, and it’s used exclusively for the inducing points. Meaning the deep network does not have any pre-training whatsoever. Moreover the inducing points are whitened at the start of training (see A.4).

---

### Official Review · AnonReviewer3 · 2020-10-29
**Extensive simulations, modest novelty compared to [Liu et.al, 2020], a key baseline that is missing**

**Rating:** 5
**Confidence:** 4

**Review:**

This paper proposes a single deep deterministic model harnessed with spectral normalization to improve uncertainty in regression and classification tasks. The method relies on deep kernel learning + inducing point approximation for GP, and spectral regularization of the deep model to avoid uncertainty collapse in OOD.

The main contribution of this paper is methodological. The paper has extensive simulations, and demonstrates the utility of the proposed approach in a wide-range of applications for both regression and classification. Having said that, the proposed approach can be seen as a modification to [Liu et.al, 2020], with different approximation and different regularization. In that sense, the novelty of the paper could be seen as modest, and a comparison with [Liu et.al, 2020] highlighting the differences in practice is missing.

More comments:

* What is the main advantage of this approach w.r.t [Liu et.al 2020]? How do these compare in terms of uncertainty estimation and computational speed? This should be included. The authors discuss [Liu et.al, 2020] in related work, highlighting the important difference that [Liu et.al, 2020] is a parametric model, but it is similar in that both formulate it as a GP and regularize for distance-awareness. A comparison with [Liu et.al, 2020] would strengthen this paper considerably.

* Figure 1 could be improved: it only shows deep ensembles as baseline, how about the other approaches discussed in the paper? Moreover, it is unclear whether vDUQ provides better in-between uncertainty compared to Deep Ensembles (similar width, but deep ensembles interpolation is more smooth.

* A deeper focus on the normalization, i.e., theoretical or empirical properties of spectral normalization and comparison with other normalization schemes would make the paper more interesting.

* The paper has some strong/categorical sentences with which we do not agree: e.g., first intro paragraph: "there is no single method that works on large datasets ..." [Liu et.al, 2020] would be such method for example.

* Simulation results are convincing in terms of utility, as the authors demonstrate that the proposed approach works in high-dimensional big datasets, and meaningful applications such as causal inference for healthcare. Yet, the experiments miss the point of elucidating how much spectral normalization compared to other normalization schemes.

* Could the authors include an ablation study showing the impact of the number of inducing points in the approximation? The authors mention as a strength that a low number of inducing points is good enough, so showing evidence for that would strengthen the paper.

---

> ### Author Response · Authors · 2020-11-17
> **Response**
>
> We thank the reviewer for taking the time to review our work and for your comments.
>
> Our work relates (Liu et.al, 2020) in the way that both draw inspiration from DUQ (van Amersfoort et al., 2020). Liu et al formalized their model as an RFF approximation, whereas we formalized ours as a method for deep kernel learning. The difference between the two is important to understand the difference in performance of the two methods.
>
> 1. The variational approximation of (Liu et.al, 2020) makes their approximation  parametric, which loses some of the guarantees on uncertainty that come with a non-parametric method. This is a very important distinction, and the main reason that most recent GP works focus exclusively on inducing point approximation over RFF. See also section 6. of Lazaro-Gredilla et al, 2010. Furthermore the RFF approximation is only tractable for RBF kernels, restricting the flexibility of the model. Specifically, RBF enforces a smoothness assumption over the functions, implying that we believe that the function that generated the data is infinitely differentiable. This need not be the case with many applications, such as financial timeseries.
> 2. SNGP (Liu et al, 2020) does not actually bound the spectral norm of the *feature extractor* because it ignores the spectral norm of batch normalization. We found that the spectral norm of BN can be very large (figure 6, appendix). We correct this and find it improves performance.
> 3. These insights allow us to outperform (Liu et.al, 2020) empirically on uncertainty as shown in Table 1, see the row called SNGP.
> 4. On your question of computational cost and speed, both SNGP and vDUQ only add minor overhead to the computational cost of a deep network (such as the WRN). SNGP computes many random features (d=1024) and vDUQ computes a small matrix inverse (10x10 for CIFAR10, which is cached at test time). So vDUQ is marginally more efficient at test time, but in practice there will be little difference on modern hardware.
>
> In Figure 1, we highlight the performance on Deep Ensembles since it’s the current state of the art in uncertainty in deep learning. The figure shows that away from the data Deep Ensembles extrapolates arbitrarily and confidently. We have updated the figure with better fit models which makes the comparison more clear.
>
> *Focus on normalization* - we agree that regularizing the spectral norm for uncertainty can benefit from more theoretical underpinning. However it is unquestionably useful in practice as established in our paper, DUQ (van Amersfoort et al., 2020) and SNGP (Liu et al., 2020). We note that in our discussion of the theoretical underpinnings we point out a flaw in the justification of [Liu et al., 2020] who claim to be enforcing the bilipschitz-ness of the mapping. This explanation is not consistent with the use of downsampling operations and cannot be the justification for its empirical performance.
>
> On *strong claims*, Liu et.al, 2020, do not demonstrate any regression experiments. We have made the sentence more precise and welcome any other examples of this type of sentence which we will correct promptly.
>
> We are curious to hear more about the “other normalization schemes” the reviewer believes would be informative to compare against, we will do our best to add these to the paper. If the reviewer refers to the comparison in section 3, then we analyze why the alternatives are not as powerful or easy to use as combining residual connections with direct spectral normalization.
>
> In the paper we report results using the minimum amount of inducing points necessary to match state-of-the-art accuracy in softmax. We ran additional experiments with more inducing points, which show similar accuracy to using 10 points but at a larger computational expense, see Appendix C. of the paper.
>
> Lázaro-Gredilla, Miguel, et al. "Sparse spectrum Gaussian process regression." The Journal of Machine Learning Research 11 (2010): 1865-1881.

---

### Official Review · AnonReviewer4 · 2020-10-29
**This paper proposed a new vDUQ (variational deterministic uncertainty quantification) that combines the inducing point GP with Deep Kernel Learning so as to obtain predictive uncertainty in deep learning for both classification and regression problems. This paper 1.	This paper is not well-written and fails to meet the ICLR standard:**

**Rating:** 2
**Confidence:** 3

**Review:**


1.	In the introduction, the author separately pointed out the issues of DUQ and DKL. However, these issues are not convincing as no citations or theoretical proof is provided in this paper. The notations in the intro are also not well-defined. X, x, x* are used without difference, which however should be clearly defined as vectors or matrices.
2.	The technical contribution is very incremental. The proposed vDUQ is simply applying the inducing point GP in the DUQ to mitigate the uncertainty collapse in DKL. The ‘’inducing point variational approximation of the GP predictive distribution’’ referred as inducing point GP is not clear for me. What exactly does ‘inducing point GP’ refer to? Why the so-called inducing point GP can speed up inference in GP model? What does ‘’decouple it from dataset size’’ mean? All these important points are not clarified in the introduction.
3.	The theoretical contributions are also not well-organized. The author fails to prove that the spectral normalization as a regularization scheme can be uses to mitigate uncertainty collapse. Moreover, how the spectral normalization guarantees the effectiveness of inducing point GP in vDUQ?
4.	I also have some concerns on the experimental results of causal inference. Why the treatment effect estimation has uncertainty settings. The authors should fully explain the uncertainty settings in causal inference, as most of the causal baselines are not proposed for uncertainty settings.

---

> ### Author Response · Authors · 2020-11-17
> **Response**
>
> We thank the reviewer for taking the time to review our work and for your comments.
>
> 1. The issues of uncertainty collapse in standard DKL are discussed extensively in Bradshaw et al., 2017. We discuss their results in paragraph 2 of the background. The limitations of DUQ (van Amersfoort et al., 2020) (DUQ is not able to do regression and it is not probabilistic) are a result of its choice of loss function and centroid updating rule and are partially discussed in the conclusion of that paper. We have clarified the notation in the text.
>
> 2. We strongly disagree that the solution is incremental. vDUQ is the first instance of DKL that matches the accuracy of a softmax model. It is also the first deterministic uncertainty quantification model in deep learning to demonstrate excellent uncertainty results on both classification and regression in a single forward pass. The fact that we built on previously introduced components does not detract from the novelty and contribution of vDUQ.
>
> We discuss the definition of an “Inducing Point GP” in the last paragraph of the introduction, and as mentioned there we closely follow the Hensman et al., 2015, formulation.
>
> Inference in an exact GP requires a matrix inverse of the size of the training data. This can be very expensive and therefore it is common to use an approximation such as the inducing point GP, which uses a smaller set of learned points that does not grow with the training data (the exact number is a hyper parameter). The matrix inverse is then of the size of the number of inducing points, which in the case of vDUQ is 10 (for CIFAR-10) compared to its training data size of 50,000. We did not discuss inducing point GPs further as the references that we point to (Hensman et al., 2015 and Bradshaw et al., 2017) do a very good job and contain great detail. We have included an algorithm box describing vDUQ for clarity.
>
> 3. The fact that we are unable to prove the benefit of spectral normalization theoretically does not detract from the fact that extensive empirical evidence, both supplied by us and by previous research,  shows that it can be very beneficial (see also van Amersfoort et al., 2020, and Liu et al., 2020). We attempt to further the theoretical advancements made in Liu et al., 2020, while pointing out the problem with dimensionality reduction in their proofs; that we point out a conceptual flaw with arguments proposed in other work should not be understood as a ‘failure’ of this paper.
>
> 4. Uncertainty quantification for causal-effect sizes has been studied as a way to get confidence intervals for individual-level effect predictions (Hill et al., 2011). We have rewritten the intro to section 5.4 to make it more clear how uncertainty quantification interacts with the effect size computation.
>
> We are very curious to hear additional examples that sparked your claim that the paper is “not well written” (which is in contrast to the comments of the other reviewers).
>
> In light of our explanations and the emphasized results in the general comment, we would like to hear if the reviewer is inclined to improve their score from the current “2”, as well as any additional concerns remaining or any points of unclarity in our response.

---

### Author Response · Authors · 2020-11-17
**General statement**

We thank the reviewers for taking the time to review our work and for their comments.

We would like to emphasize some of the contributions we present in this paper which span multiple communities:
1. In Bayesian Deep Learning, we develop a new methodology which obtains state-of-the-art out-of-distribution detection for a single model, outperforming not just DUQ and SNGP, but also a standard softmax model and previous approaches in DKL.
2. In contrast to existing methods (including DUQ and SNGP), vDUQ is demonstrated to produce high quality uncertainty both on classification, as well as *regression problems* (which to the best of our knowledge were not studied before in the context of deterministic uncertainty quantification in deep learning).
3. In Gaussian processes and Deep Kernel Learning, we demonstrate that using an interdomain approximation and placing the inducing points in *feature space* can match state of the art accuracy on softmax with a Wide ResNet (96.2%). To the best of our knowledge, prior works place inducing points mostly in input space or spectrum space. We find this very exciting - closing the gap between the GP literature and deep learning performance is a big sought-after endeavour (see detailed discussion in section “5.3 CIFAR-10 AND SVHN”, contrasting to existing GP approaches). For reference, even previous DKL approaches such as SV-DKL, which aim to scale GPs to large high dimensional data, obtain ~3% below softmax accuracy for the same network.
4. In Causality, we give new state of the art results on standard benchmarks in causal-effect prediction using medical data, a real-world regression application where uncertainty is of utmost importance.

Based on the feedback, we have made the following changes to the paper:
- Added an ablation study on the inducing points, showing that using more inducing points does not lead to better performance (Appendix C.)
- Added an algorithm box describing vDUQ. Our method can be trained end-to-end with simple gradient descent and off-the-shelf GPyTorch + PyTorch modules.
- We have rewritten the methods section to clarify the context of the bi-lipschitz constraint. We acknowledge that this caused confusion with the reviewers, and have now moved the theoretical discussion of limitations to a new section called “Limitations”.
- We have updated figure 1 with better fit models which alleviates some of the concerns by the reviewers.

These changes are already visible by looking at the paper link.

---

### Decision · Program_Chairs · 2021-01-07
**Final Decision**

**Decision:**

Reject

**Comment:**

The reviewers all agreed that the paper represent thorough work but also is closely related to existing literature. (All referees point to other non-overlapping literature so it is a crowded field the authors have entered.) The amount of novelty (needed) can always be discussed but given the referees unanimous opinion and knowledgable input it is better for this work to be rejected for this conference. Using this input can make this work a good paper for submission elsewhere.